# Characteristics of Sunburn Browning Fruit and Rootstock-Dependent Damage-Free Yield of Ambrosia™ Apple after Sustained Summer Heat Events

**DOI:** 10.3390/plants11091201

**Published:** 2022-04-29

**Authors:** Hao Xu, Yoichiro Watanabe, Danielle Ediger, Xiaotang Yang, Davis Iritani

**Affiliations:** Summerland Research and Development Centre, Agriculture and Agri-Food Canada, Summerland, BC V0H 1Z0, Canada; yoichiro.watanabe@agr.gc.ca (Y.W.); danielle.ediger@agr.gc.ca (D.E.); xiaotang.yang@agr.gc.ca (X.Y.); naoki.iritani@agr.gc.ca (D.I.)

**Keywords:** delta absorbance *I*_AD_, epidermal cell integrity, firmness retention, fruit dry matter, fruit quality, heat stress, rootstock vigor, skin spectral characteristics, tissue water potential

## Abstract

The 2021 summer heat waves experienced in the Pacific Northwest led to considerable fruit damage in many apple production zones. Sunburn browning (SB) was a particularly evident symptom. To understand the mechanism underlying the damage and to facilitate the early assessment of compromised fruit quality, we conducted a study on external characteristics and internal quality attributes of SB ‘Ambrosia’ apple (*Malus domestica* var. Ambrosia) and evaluated the fruit loss on five rootstocks. The cell integrity of the epidermal and hypodermal layers of fruit skins in the SB patch was compromised. Specifically, the number of chloroplasts and anthocyanin decreased in damaged cells, while autofluorescent stress-related compounds accumulated in dead cells. Consequently, the affected sun-exposed skin demonstrated a significant increase in differential absorbance between 670 nm and 720 nm, measured using a handheld apple DA meter, highlighting the potential of using this method as a non-destructive early indicator for sunburn damage. Sunburn browning eventually led to lower fruit weight, an increase in average dry matter content, soluble solids content, acidity, deteriorated weight retention, quicker loss of firmness, and accelerated ethylene emission during ripening. Significant inconsistency was found between the sun-exposed and shaded sides in SB apples regarding dry matter content, firmness, and tissue water potential, which implied preharvest water deficit in damaged tissues and the risk of quicker decline of postharvest quality. Geneva 935 (G.935), a large-dwarfing rootstock with more vigor and higher water transport capacity, led to a lower ratio of heat-damaged fruits and a higher yield of disorder-free fruits, suggesting rootstock selection as a long-term horticultural measure to mitigate summer heat stress.

## 1. Introduction

In recent years, the increased frequency of sustained summer heat waves with extreme maximum temperatures has become an intensifying concern to many crop production zones globally. Tree fruit crops, which have perennial life cycles and long growing seasons, are especially vulnerable to such heat events. Sustained heat above 35 °C in the middle of a growing season can impose multifaceted negative impacts on apple fruit development. Firstly, it inhibits leaf photosynthesis and accelerates respiration, leading to a decrease in net photosynthetic gain and the alteration in carbohydrate allocation [1]. Consequently, less sucrose and sorbitol are translocated to fruits [2]. During warm nights, leaves may mobilize starch from fruits to support high leaf respiration demand, a phenomenon that is often referred to as starch dilution and can complicate the interpretation of fruit starch index [3]. Fruit at harvest may have less total soluble solids, lower total dry matter per fruit, and smaller size [2,3,4]. Secondly, heat can decrease water and mineral supply to fruits due to intensified leaf–fruit competition [5]. During the fruit expansion stage, these decreases can lead to smaller fruits with less water content; despite the deceased carbohydrates per fruit, the soluble solids content % and dry matter content % values may increase.

The compromised supply of carbohydrates, mineral nutrients, and water to fruit can result in localized water loss, altered epidermal cell structure, and cell rupture. It can also hinder vascular tissue development and disrupt cellular metabolism due to increased peroxidative damage to macromolecules. These cellular-level damages can lead to variable preharvest physiological disorders, such as shriveled, deformed, or asymmetrical fruits, and higher incidents of lenticel marking, cracking, russeting, bitter pit, and water core; loss of turgor and lignification in damaged and dead cells may lead to a reduction in elasticity of the affected tissue and an increase in fruit firmness reading [5,6].

Direct external damage can occur when fruit surface temperature or radiation reaches certain thresholds, overexposing the cells within and under the fruit skin. When fruit surface temperature rises above 45 °C, the resulting death of skin cells leads to necrosis. Sudden exposure to radiation causes photooxidative sunburn and localized loss of pigments, leaving a bleached patch where it used to be shaded; this is usually seen when hot and sunny days concur with strong wind or summer pruning. Sunburn browning (SB) is the third type of sunburn damage, due to the combinative effects of excessive UV-B radiation and high fruit surface temperature. It is often associated with chlorophyll degradation, wax melting in cuticles, and deposition of stress response compounds, giving the damaged patch a brownish or purplish-red appearance [6,7,8,9]. Though necrotic and bleached fruits are usually discarded during sorting, fruits with mild and moderate SB, if not identified at harvest, can develop significant delayed scald and stain during storage.

In summer 2021, many apple production zones in the Northern Hemisphere were challenged by extreme heat events. In late June and July, Interior British Columbia, Canada, experienced several heat waves with record-breaking maximum temperature and prolonged duration, leading to considerable heat damage to apple fruits during the stage of fruit expansion. Sunburn browning (SB) was particularly noticeable on the southwest side of the well sun-exposed Tall Spindle Axe (TSA) canopy in high-density planting systems. The production of Ambrosia™ apple (*Malus domestica* var. Ambrosia, ‘Ambrosia’), a variety that has been gaining popularity worldwide because of its highly appraised bicolor and delicacy, great storage quality, and good economic return, was heavily affected. This provoked the realization that, without knowing the cultivar’s responses to heat, it would be difficult to accurately predict or precisely assess the impacts of heat and other associated abiotic stress. Therefore, to be able to design successful mitigation measures, research on this cultivar is needed to understand the mechanism underlying SB symptoms. Such insight would guide the development of early assessment techniques for the detection of compromised fruit quality, as well as exploration of potential long-term mitigation measures. With these goals in mind, we characterized the external and internal attributes of the SB apple with reference to damage-free apples, including skin cuticle and epidermal anatomy, skin delta absorbance *I*_AD 670–720 nm_ and surface temperature, fruit weight at harvest and postharvest weight retention, dry matter content, soluble solids content, acidity, firmness, ethylene emission, and tissue water potential. The inconsistency between the sun-exposed and the shaded sides in selected traits was examined in SB apples. In a rootstock trial at the experimental farm of Summerland Research and Development Centre, Agriculture and Agri-Food Canada, heat damage ratio and damage-free fruit yield were evaluated in five rootstocks of different vigor, i.e., large-dwarfing Geneva 935 (G.935) and Geneva 202 (G.202), moderate-dwarfing Malling 26 (M.26) and Malling 9NIC29 (M.9), and small-dwarfing Budagovsky 9 (B.9) [10]. Furthermore, we studied the implications for preharvest stress and postharvest quality. The findings of this study improve our understanding of apple fruit physiology that underlines SB symptoms and facilitate decision making on fruit sorting and rootstock selection of ‘Ambrosia’ and similar varieties, to cope with a warmer future.

## 2. Results

The apple production zone in the Okanagan Valley of Interior British Columbia experienced a dry and hot growing season in 2021. In Summerland, the moisture deficit from 1 May to 30 September 2021 was 656 mm, which was 35 mm more than the historical average measured since the 1960s. The T-sum (accumulated mean daily temperatures above 0 °C) of May to September was 3008.85 °C, 293 °C higher than the 60-year average. In late June, July, and mid-August, daily maximum and minimum temperatures exceeded the 2010–2020 average on a few continuous days (Figure 1A). On 28–30 June, the maximum temperature was above 40 °C.

Immediately after this heat wave, necrotic and slight-to-severe SB apples were spotted on the southwestern side of the canopy, regardless of rootstocks or the tree’s location in the trial. The SB manifested on the sun-exposed side of the fruit, and the symptoms varied from yellowish patches in mildly damaged apples to red-purple patches on moderately and severely damaged fruits in early July (Figure 1B). As fruit expanded and matured, the symptoms on moderate-to-severe SB apples deteriorated into brownish patches, with a hardened and roughened surface texture, whereas slight-to-mild damages became less noticeable as fruit color developed (Figure 1C).

### 2.1. Cellular Structures of Fruit Skin in Sunburn Browning Apple

Due to their autofluorescent nature, many plant compounds can easily be observed using a confocal microscope, with little to no sample preparation. With this in mind, we sought to characterize any changes in chlorophyll and anthocyanin content at the subcellular level in apple SB damaged skin cells. Chlorophyll autofluorescence within chloroplasts (Figure 2, Cl and insert) were clearly seen within the green fluorescence channel of undamaged cells and to a lesser extent in moderately damaged living cells. Anthocyanin autofluorescence within vacuoles (V) was seen in the red fluorescence channel, in both undamaged cells and living cells, in moderately damaged tissue. In severely damaged or dead cells, no characteristic chlorophyll or anthocyanin signal was observed. Interestingly, these collapsed dead cells in damaged tissue accumulated an autofluorescent compound or compound conjugate that exhibited intense fluorescence in both the green and red channels. Though not identified in this study, this compound displays fluorescence characteristics similar to many stress-related compounds.

Transverse sections of ‘Ambrosia’ apple peels further revealed that in moderate and severe SB damage tissue, the death and collapse of cells occurred several layers deep into the subepidermal cell layer (Figure 3A). These collapsed cells were also filled with the highly autofluorescent compound(s) that displayed similar autofluorescence characteristics to those of the overlying true cuticle (Figure 3A). The collapse of cells and the accumulation of autofluorescent compound(s) made it difficult to discern between the cuticle (Cu) and epidermal (Ep)/subepidermal (Se) cell layers. Thus, in this study, the resulting structure was described as an abnormal cuticular layer composed of both true cuticle and collapsed cells. The thickness of this ‘cuticle’ layer appeared to increase in thickness with increasing severity of sunburn damage due to the integration of more layers of collapsed and dead cells being filled with the autofluorescent compound(s) (Figure 3A,B).

### 2.2. Spectral Characteristics of Fruit Skin in SB Apple

Two weeks prior to harvest, delta absorbance *I*_AD 670__–__720 nm_ (*I*_AD_) was found to be not significantly different between the sun-exposed side and the shaded side on normal fruit (Figure 4A). The *I*_AD_ of the shaded side of SB fruit was similar to that of normal fruit, whereas the value was significantly higher on the sun-exposed browning side, compared with the shaded side of the same fruit and sun-exposed side of normal fruit (Figure 4A).

Maximum (T_max_), Minimum (T_min_), mean (T_mean_), and differential surface temperature (T_diff_) on the sun-exposed side were compared between normal and SB fruits in late August. The SB fruit had lower T_min_ and T_mean_, and larger T_diff_, compared with the normal fruit; T_max_ was similar between the two types of fruits (Figure 4B).

### 2.3. Dynamics of Internal Attributes of SB Apple

At harvest, SB apple had significantly lower fruit weight but higher soluble solids content % (SSC%) and malic acid concentration (Table 1). During weeks 1–3 of ripening at room temperature, the weight of SB fruits dropped faster; by the end of the 3rd week, average weight retention was about 93% in SB fruits, which was 2–3% lower than normal fruits (Figure 5A).

SB apple had higher overall fruit firmness at harvest, followed by a quicker loss of firmness during ripening (Figure 5B). After air storage for two months at 4 °C and for three weeks at 20 °C, the firmness of SB apple dropped to 77.1% and 60.8%, respectively, whereas in normal fruits, firmness declined to a lesser extent as 87.1% and 75.3% (Figure 5B). There was inconsistent flesh firmness within the SB fruit, with significantly higher firmness on the damaged sun-exposed side than on the shaded side (Table 1). In contrast, the two sides in normal fruits had similar firmness, which was lower than their corresponding sides in SB fruits (Table 1).

Significant inconsistency was also observed in dry matter content % (DMC%) and tissue water potential of fruit hypanthium (*Ψ*_fruit_) within the SB apple. Its sun-exposed side had higher DMC% and lower *Ψ*_fruit_ than its shaded side. DMC% and *Ψ*_fruit_ of the SB sun-exposed side were about 5% higher and 0.7 MPa lower than those of the normal sun-exposed side, respectively (Table 1).

Changes in internal ethylene production during air storage were measured in normal and SB fruits (Figure 6). At 20 °C, internal ethylene concentration in normal fruits remained low (<1 µL/L) until Day 31 and then increased gradually. The onset of climacteric increase in ethylene production occurred much earlier in SB fruits and progressed more rapidly, with significantly higher concentration throughout the evaluation period (*p* ≤ 0.05).

### 2.4. Vigor, Yield, and Ratio of Heat-Damaged Fruits on Five Rootstocks

The rootstocks demonstrated differences in vigor as expected. The trunk cross-sectional area (TCSA) by the end of the 2021 growing season was 7.22 ± 0.35 cm^2^ in the G.935, 7.21 ± 0.40 cm^2^ in G.202; 5.50 ± 0.41 cm^2^ in M.9, 5.38 ± 0.25 cm^2^ in M.26, and 4.14 ± 0.22 cm^2^ in B.9 (mean ± standard error; data not graphed). More vigorous rootstocks resulted in a higher yield per tree, with large-dwarfing G.935 being the most productive, followed by large-dwarfing G.202 and moderate-dwarfing M.26 and M.9; ‘Ambrosia’ on small-dwarfing B.9 had the lowest yield per tree (Figure 7A). The ratio of heat-damaged fruits was on average 5% in G.935, significantly lower than that in the smaller-dwarfing rootstocks M.26, M.9, and B.9 (Figure 7B). B.9 and G.935 excelled the other three rootstocks in the projected yield of damage-free fruits per acre based on the estimated optimum planting density for each rootstock (Figure 7C). G.935 led to the highest average fruit weight (Figure 7D).

## 3. Discussion

### 3.1. Fruit Responses to Heat and UV Stress

Apple fruit has a set of pre-formed mechanisms to avoid stresses that are associated with excessive temperature and UV radiation. As the first barrier against the external environment, the hydrophobic cuticle layer on the fruit skin surface can effectively reduce water loss [11]. In this study, due to the integration of damaged epidermal and subepidermal cell layers with the true cuticle, increased thickness of the merged cuticular layer was observed on the sun-exposed side of SB apples (Figure 3A). Further, the thickness also increased with the increased severity of SB (Figure 3B). This positive correlation suggests the role of cuticle waxes or thickened cuticles in preventing water loss from the wounded skin patch. Attributed to their optical properties and high specific heat capacity, cuticles can also provide UV protection [12] and contribute to the stabilization of surface temperature under variation in ambient temperature [13]. Increased cuticle thickness (Figure 3B) and lower T_min_ (Figure 4B) were observed in the sun-exposed skin of SB apples, implying the role of cuticle layer modification in exerting a localized cooling effect on the fruit surface. Lower T_min_ contributed to a larger T_diff_ on the SB side. This is inconsistent with the decrease in temperature difference and the increase in crop water stress index with the intensification of heat and water stress reported in ‘Granny Smith’ apples [14]. It suggests that cultivar-specific fruit skin characteristics and environmental conditions during infrared imaging should be taken into consideration, to improve the accuracy of surface-temperature-based detection methods.

The presence of plant pigments in the epidermis, such as chlorophyll, anthocyanin, and carotenoids, can increase light absorption by the fruit and contribute to harnessing excessive light energy [15]. Chlorophyll and anthocyanin degradation due to SB was obvious in the subepidermal layer; under moderate SB, anthocyanin showed slightly more stability than chlorophyll (Figure 2, central micrograph). The autofluorescent substance that filled the dead cell under severe SB is speculated to be stress-related, either as a product of oxidation such as quinone, which autofluoresces in both green and red channels [16], or, as a stress-responsive phenolic compound or compound conjugate to prevent further damage to the surrounding tissues. The hydrophobic polymers—lignin and suberin—have been reported in russeting apple skin as a response to environmental stresses [17], which gave the disordered fruit a brownish color and rough texture that resembled the SB patch. The increased abundance of quercetin, a flavonoid glycoside, as well as a yellowish-brown pigment, was reported in mild SB ‘Fuji’ apples along with the degradation of chlorophyll and anthocyanin, possibly contributing to the manifested appearance of the yellowish patch [7,8]. The concentration of chlorogenic acid, another phenolic compound functioning as an intermediate in lignin biosynthesis, was found to increase in moderate SB apples of four red and yellow varieties [9]. Given the wide possibilities of the nature of the autofluorescent compound(s) accumulating in SB damaged cells, further research to identify these compounds is needed. Histological stains such as Congo red and safranin can be used to quickly identify whether these compounds are phenolic in nature [18]. High-performance liquid chromatography (HPLC) can then be used to identify these compounds based on their retention time [7,8], and to quantify the accumulation of these compounds throughout fruit development and storage. Future studies on the profiling and interplay of these compounds, some of which share transcription factors and biosynthesis pathways, would help to decipher fruit responses to stresses at the cellular level.

### 3.2. Heat, Water, and Rootstock Mitigation

Heat and drought stress often aggravate one another in water-limited environments. Therefore, to some extent, improving plant–water relations can alleviate heat stress. In this vein, overhead irrigation cooling, protective netting over the canopy, and protectant sprays containing carnauba wax, modified clay, and calcium carbonate, are proven to be effective to prevent or reduce apple sunburn; these measures either directly provide evapotranspiration cooling or reduce the crop water demand [19]. In this study, the use of vigorous rootstocks was shown as another tool to reduce apple SB. Larger rootstocks have some hydraulic advantages in the water transport system, including larger trunk cross-sectional area, more actively transporting xylem, larger root volume to exploit wider and deeper soil profiles, and larger root xylem vessel elements [20,21,22,23]. These traits give rootstocks higher water transport capacity to meet the scion water demand; when irrigation is sufficient, the better water supply of larger rootstocks could contribute to more transpirational cooling in the canopy and help to alleviate heat stress. In addition, higher foliage density on larger rootstocks can provide more shading to the fruit, reducing surface temperature and direct UV exposure. G.935 rootstock excelled over other rootstocks in yield of sunburn-free fruits (Figure 7C) and fruit weight (Figure 7D), exhibiting its potential in producing high-quality ‘Ambrosia’ fruits in a warmer future. In other words, without other preventive measures, this rootstock could render ‘Ambrosia’ apples less susceptible to heat stress and keep the sunburn damage ratio under 5%. Continued study on its interaction with ‘Ambrosia’ apples, as well as its combination with other apple varieties, would facilitate its evaluation as a long-term measure to mitigate heat.

Conversely, the smallest rootstock B.9 led to the highest damage ratio (Figure 7B). However, its small TCSA allows for 2180 trees per acre, i.e., 1000 more trees per acre than the optimum planting density of the large-dwarfing rootstocks. Due to its higher optimum planting density, its projected damage-free fruit yield per acre was comparable to G.935 (Figure 7C). As its less vigor indicated lower water demand, B.9 could be more adaptable to the orchard environment where water supply is a constantly limiting factor.

### 3.3. Fruit Quality Deterioration in SB Apple

Daily maximum and minimum temperatures suggested both warmer days and nights on a few continuous days in June and July 2021 when compared with the 2010–2020 average (Figure 1A). This would cause a reduction in net photosynthetic gain due to impaired photosynthesis and accelerated respiration; consequently, fewer carbohydrates would be partitioned to fruits. In this study, SB ‘Ambrosia’ apples had lower fruit weight (Figure 5A, Table 1), which is consistent with previous reports on reduced fruit size due to elevated summer temperatures [2–4.6]. Elevated SSC% and DMC%, and lower water content %, as interpreted in the context of lower tissue water potential (Table 1), implied water deficit in SB apples at the preharvest stage. Despite the higher percentage values, the abundance of soluble solids and dry matter per fruit was lower in SB apples, due to their lower fruit weight.

The result of changes in internal ethylene concentration of ‘Ambrosia’ apples during air storage (Figure 6) was in line with the findings in the ‘Granny Smith’ variety, in which significantly higher levels of ethylene were found in sun-injured apples during cold storage [24]. While ethylene is critical in the normal ripening process in climacteric fruits, it is also strongly associated with senescence and responses to numerous abiotic stresses and mechanical injury [25]. The results indicated that excessive solar radiation and high fruit surface temperature triggered a stress response that was modulated by ethylene in apples.

In addition, significant inconsistencies in skin *I*_AD_ (Figure 4A), flesh firmness, DMC%, and water potential (Table 1) were found between sun-exposed and shaded sides in SB apples. Higher firmness was observed on the damaged sun-exposed side, likely due to collapsed dead cells or accumulation of stress-related compounds that increased the rigidity of the sun-exposed damaged tissue. High DMC% and low water potential on the SB side pointed to tissue water deficit; in addition, normal firmness, DMC%, and water potential on the shaded side of SB apples indicated the damage was localized to the sun-exposed side. These inconsistencies could cause structural and osmotic imbalance within the fruit, leading to quicker deterioration of fruit quality, such as worsened retention of fruit weight and firmness during ripening (Figure 5). The increased decay incident in the affected skin patch was also observed in SB apples during ripening. Subtle cracking in the wax of the cuticle layer due to heat stress could make the fruit more susceptible to pathogens, cause faster fruit dehydration during ripening, and subsequently shorten shelf life [13,26]. Understanding these comprehensive heat impacts on fruit quality, storage ability, and shelf life would facilitate the practitioners’ decision making on selective picking, fruit processing, and strategic marketing.

### 3.4. Detection Using Handheld Tools

Although the cuticle, epidermis, and spectral characteristics of the fruit surface were altered due to SB, the visual symptoms can be masked off in red cultivar during coloration and maturation, making the preharvest diagnostics difficult. Affordable handheld instruments that can reliably distinguish SB symptoms at the preharvest stage can assist growers with early detection and evaluation of the damage.

In this study, three handheld, non-destructive instruments were tested. The alterations in cuticular and epidermal structures, as well as the presence of the stress response compounds, changed the differential absorbance at 670–720 nm and the heat diffusion capacity of the affected SB skin patch. This range overlapped with 715–750 nm, one of the three spectral ranges that were identified as responsible for the prediction of sun injury development in ‘Granny Smith’ apples [27]. Although it is often used to estimate chlorophyll degradation during fruit maturation, the apple DA meter was able to detect the increased *I*_AD_ on the SB side with statistical significance, with a measurement time in seconds (Figure 4A). Lower T_min_ and T_mean_, and larger T_diff_, were also detected using an infrared imager (Figure 4B); however, the method involves data processing, which is time-consuming. Significantly higher DMC% was detected on the SB side in less than a minute using Felix-750 m; the advantage was its clear distinction between sun-exposed and shaded sides in SB apples, as well as between the SB and the normal sun-exposed sides (Table 1); the limitations were its higher cost and a sophisticated calibration procedure of the instrument.

### 3.5. Summary and Future Perspectives

Sunburn browning deteriorated multiple critical quality attributes of ‘Ambrosia’ apples after the sustained heat events in the summer of 2021, leading to lower fruit weight, deteriorated weight retention, quicker loss of firmness, and accelerated ethylene emission during ripening. The increased average DMC% and SSC%, and the decreased fruit weight and tissue water potential converged to indicate water deficit Inconsistent *I*_AD_, DMC%, firmness, and water potential within an SB fruit underlay its compromised storage ability and shelf life. Future studies should include the evaluation of SB-related progression of skin red-color development, fruit maturation, ripening, as well as postharvest disorders such as internal browning due to trapped CO_2_ while harvested on warm days and water core development due to disrupted sorbitol transport.

Compared with an infrared imager and Felix-750 m, the DA meter was shown to be a practical tool to detect moderate SB in ‘Ambrosia’ apples. Its effectiveness awaits to be tested in a larger sample pool that consists of a range of SB severities and apple varieties. In addition, a better understanding of cellular structures and the interactions of stress-responsive compounds in SB fruit skin can improve the accuracy of spectral detection of subtle and mild SB symptoms.

The quantity of SB-damaged fruit and the reduced fruit weight suggested the impact of heat stress on carbohydrate partitioning in fruits. To elucidate the underlying mechanism, it is necessary to quantify the reduction in source strength and examine the potential damage to vascular tissues in stem and fruit petiole under heat. Whether the sustained heat has altered the partitioning to other sinks such as shoot extension, bud development, trunk increment, and root growth remains to be investigated.

The large-dwarfing rootstock G.935 led to a lower ratio of SB fruit, higher damage-free yield, and larger fruit in the third leaf planting after a summer of unprecedented heat. To ensure the robustness of the results obtained from the one-year data, the investigation on the performance and susceptibility of G.935 should be extended to the later stage of the planting, under heat stress and other concurrent environmental extremes such as excessive UV radiation or water deficit. A long-term, multi-location evaluation of vigorous rootstocks that possess higher water transport capacity and higher foliage density can generate valuable information for rootstock selection to mitigate heat stress.

## 4. Materials and Methods

### 4.1. Experimental Design and Study Materials

The study trial was located at the experimental farm of Summerland Research and Development Centre, Agriculture and Agri-Food Canada (49°33045″ N, 119°38055″ W, elevation 454 m). In the third leaf planting, ‘Ambrosia’ trees on Budagovsky 9 (B.9), Malling 9NIC29 (M.9), Malling 26 (M.26), Geneva 202 (G.202), and Geneva 935 (G.935) were grown in silt–loam soil in a complete randomized block design (*n* = 6 plots for each rootstock, 3 trees per plot, 18 trees per rootstock). Trees were trained to Tall Spindle Axe structure in 3′ × 12′ (0.91 m × 3.66 m) high-density planting. After June drop, fruitlets were thinned to one king fruit per spur to attain a moderate crop load level of 6.5–7.5 fruits per cm^2^ of trunk cross-sectional area at harvest. Irrigation was applied through drip lines from 8 AM to 10 AM every three days, from May to early October. Overhead irrigation for cooling was off in late June–early July, and on for 5 min at 12 PM and 3 PM each daily, from mid-July to early September. Weather data were acquired from farmwest.com (Environmental Canada, Summerland weather station (South Okanagan, British Columbia); accessed on 12 March 2022).

In mid-August, 100 fruits with moderate SB and 100 damage-free normal fruits in their adjacency were tagged in the trial. Fruits were positioned at about 1.5–1.8 m high on the southwestern side of the canopy; sampling was randomized across the rootstocks and the tree locations in the trial. The tagged fruits and additional 100 fruits per type were harvested on 20–23 September for laboratory analysis.

### 4.2. Spectral Characteristics of Fruit Skin

In early September, two weeks prior to harvest, delta absorbance *I*_AD 670–720 nm_ (*I*_AD_) was non-destructively measured on sun-exposed and shaded sides of the tagged fruits, using a handheld Apple DA meter (Sinteleia, Bologna, Italy) (*n* = 100). Maximum (T_max_) and minimum (T_min_) skin surface temperatures on the sun-exposed side were detected on 26 August 2021, using FLIR E8 Infrared camera (FLIR^®^ Systems Inc., Wilsonville, OR, USA; 7 mm focal lens, 320 × 240 IR resolution; software FLIR Tools V. 6.4) (*n* = 100); mean (T_mean_) and differential surface temperatures T_diff_ were calculated as average and discrepancy of T_max_ and T_min_ (T_max_ − T_min_), respectively.

### 4.3. Confocal Microscopy

Fresh apple skin peels were hand-sectioned using a double edge razor blade and mounted onto a 35 mm diameter glass-bottom Petri dish (Electron Microscopy Sciences, Hatfield, PA, USA). Sections were immersed in distilled water and weighed down by an aluminum stub and immediately imaged. Samples were imaged on a Leica Microsystem CMS GmbH (Wetzlar, Germany) TCS SP8 white light laser confocal system with a 25 × 0.95 NA water immersion objective. Chlorophyll autofluorescence was imaged using a 488 nm laser and 490–568 nm emission filter (green fluorescence channel), while anthocyanin autofluorescence was imaged using a 577 nm laser and 582–720 nm channel (red fluorescence channel). Cuticle thickness was measured from the outer edge of the cuticle to the closet edge of the first living cell layer, identified by the presence of plastids and/or vacuoles.

### 4.4. Fruit Compositional Attributes

Fruit weight was measured using a compact bench scale (Ohaus R71MHD35 Ranger 7000, Parsippany, NJ, USA) at harvest (*n* = 100), and then after one week, two weeks, and three weeks of ripening at 20 °C for weight retention evaluation (*n* = 20).

At harvest, soluble solids content (SSC %) and acidity (estimated malic acid concentration, in mg/100 mL) were evaluated (*n* = 6 juice samples, each consisting of juice from 5 fruits, 4 slices per fruit). SSC was measured using a portable refractometer (30P, Mettler Toledo, Columbus, OH, USA). The malic acid concentration was estimated by acid–base end-point titration, using OrionStar T940 titrator (Thermo Scientific, Waltham, MA, USA).

On both sun-exposed and shaded sides, fruit dry matter content (DMC%) was estimated using a Felix-750 Produce Quality Meter (Felix Instruments Inc., Camas, WA, USA) (*n* = 100); tissue water potential of fruit hypanthium (*Ψ*_fruit_) was measured using a WP4C potentiometer (Meter Environment) (*n* = 6) [10]; fruit firmness was measured using a Fruit Texture Analyzer (FTA-G25, GÜSS Manufacturing Ltd., Strand, South Africa), at harvest (*n* = 30), after two months of air storage at 4 °C (*n* = 20), and after three weeks of ripening at 20 °C following the cold storage (*n* = 20). The inconsistencies between sun-exposed and shaded sides were examined in terms of these compositional attributes.

### 4.5. Ethylene Emission

The ‘Ambrosia’ fruits with and without SB were harvested, and 15 fruits for each group were selected for the measurement. Average maturity based on DA meter readings was between 0.65 and 0.74 at harvest. Fruits were then stored at 20 °C in air storage during the evaluation. Internal ethylene concentrations were determined using a gas chromatograph (Model 8610 C, SRI Instruments, Torrance, CA, USA), fitted with a 3 m × 2.1 mm i.d. stainless steel column packed with >0.254 mm and <0.318 mm particle diameter Hayesep D (Supelco, Oakville, ON, Canada). Afterward, 1 mL of gas from the core of the fruit was taken and then injected into the instrument.

### 4.6. Rootstock Evaluation

Yield, the ratio of sunburn damaged apples in total fruit counts per tree, and trunk cross-sectional area (TCSA) were recorded for the five rootstocks (*n* = 6 plots per rootstock, 3 trees per plot). Damage-free fruit yield per tree was calculated as yield multiplied by the ratio of damage-free fruits. Yield per acre was projected according to the tree counts per acre at the estimated vigor-specific optimum planting density, based on the cumulative 10-year TCSA per acre of each rootstock vigor class equalized to that of the reference M.9 at its standard 3′ × 11′ planting density (0.91 m × 3.35 m) [28,29]. Planting density and tree counts per acre used in the calculation were 2′ × 10′ (0.61 m × 3.05 m) and 2180 trees for B.9; 3′ × 11′ (0.91 m × 3.35 m) and 1320 trees for M.9; 3.1′ × 12′ (0.94 m × 3.66 m) and 1180 trees for M.26 and G.935; and 3.9′ × 12′ (1.19 m × 3.66 m) and 930 trees for G.202, respectively. The average fruit weight of each rootstock was measured at harvest (*n* = 6 plots per rootstock, 3 trees per plot, 5 fruits per tree).

### 4.7. Statistical Analysis

Statistical significance analysis (ANOVA, Tukey Pairwise Comparisons) and graphing were conducted in OriginPro 8.0 (OriginLab, Northampton, MA, USA). Lines in box plots from bottom to top represent the minimum, lower quartile, median, upper quartile, and maximum values. Dots with error bars are the mean ± the standard error; different letters in the same table column and in the same figure subpanel indicate significant differences (*p* ≤ 0.05) unless stated otherwise.

## Figures and Tables

**Figure 1 plants-11-01201-f001:**
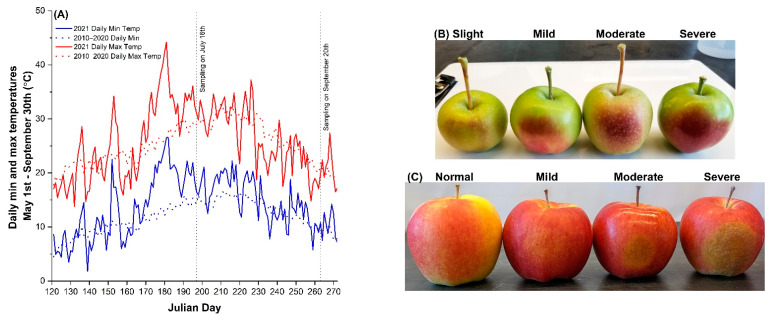
Sunburn browning (SB) of ‘Ambrosia’ apple due to sustained heat in the summer of 2021 at the Summerland experimental farm, the Okanagan Valley, British Columbia, Canada: (**A**) the daily maximum and minimum air temperatures in May to September of 2021 and 2010–2020 average; (**B**) the severity of sunburn browning symptoms in ‘Ambrosia’ apple fruit sampled on 16 July (Julian Day 197); (**C**) the progression of SB symptoms in mature apples sampled on 20 September (Julian Day 263).

**Figure 2 plants-11-01201-f002:**
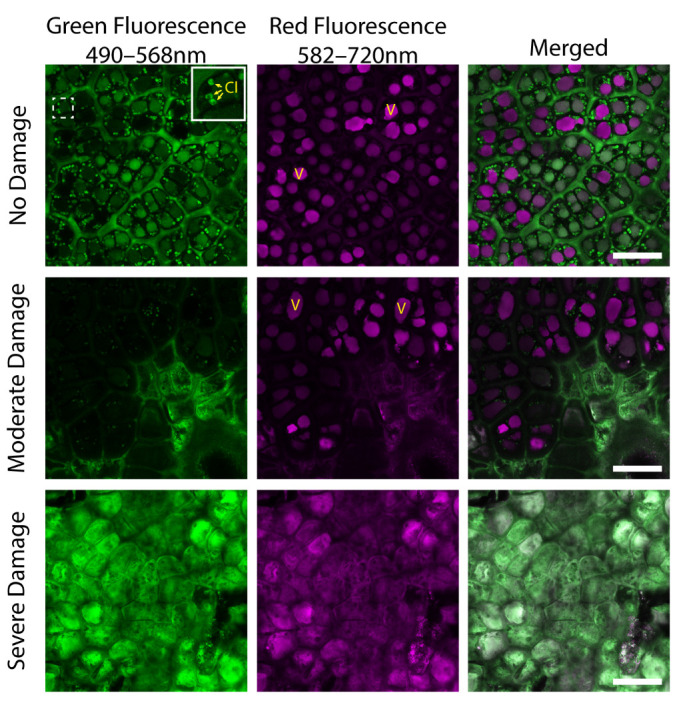
Sunburn damage results in decreased chlorophyll and anthocyanin autofluorescence and the eventual death of epidermal cells which become filled with an autofluorescent stress-related compound. Confocal micrographs of tangential sections of ‘Ambrosia’ apple epidermis exhibit no, moderate, and severe sunburn damage. V, vacuole; Cl, chloroplasts. Scale bars = 50 µm.

**Figure 3 plants-11-01201-f003:**
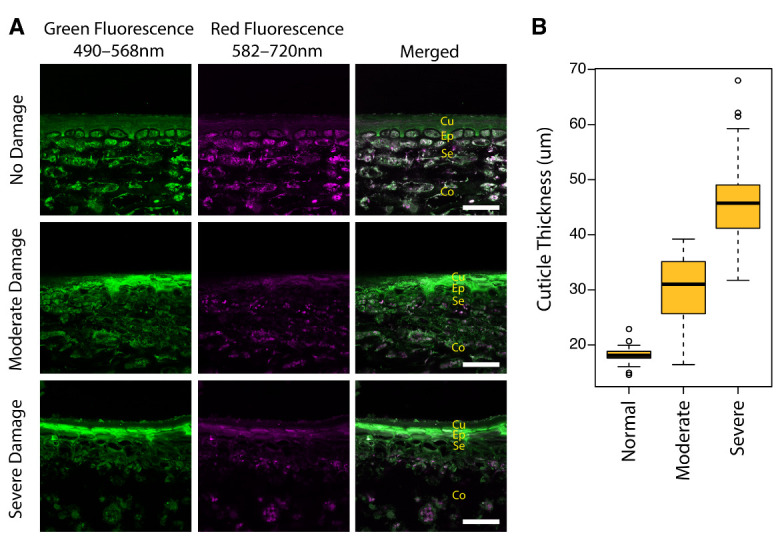
Sunburn damage results in the death and collapse of several layers of subepidermal cells and the increased thickness of the cuticular layer: (**A**) confocal micrographs of transverse sections of ‘Ambrosia’ apple peels exhibiting no, moderate, and severe sunburn browning damage. Cu, cuticle; Ep, epidermal tissue; Se, subepidermal tissue; Co, cortex. Scale bars = 50 µm; (**B**) Boxplot of measurements of cuticular thickness in the sun-exposed skin with no, moderate, and severe sunburn browning damage, as measured from the edge of the cuticle to the first living cell layer (*n* = 50 measurements from 10 micrographs each; circles represent outliers).

**Figure 4 plants-11-01201-f004:**
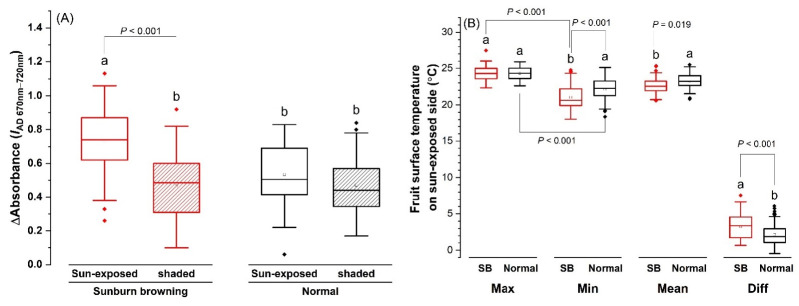
Fruit skin spectral characteristics of normal and sunburn browning (SB) ‘Ambrosia’ apple measured prior to harvest in 2021: (**A**) delta absorbance *I*_AD 670__–__720 nm_ of sun-exposed and shaded sides measured two weeks prior to harvest; (**B**) maximum (T_max_), minimum (T_min_), mean (T_mean_), and differential (T_diff_) skin surface temperatures of sun-exposed side measured in late August. Different letters a and b in each pair of SB-normal comparisons stand for statistical significance at *p* ≤ 0.05 (ANOVA); statistical significance between T_max_ and T_min_ of the same fruit type was shown in brackets (ANOVA, Tukey pairwise comparison).

**Figure 5 plants-11-01201-f005:**
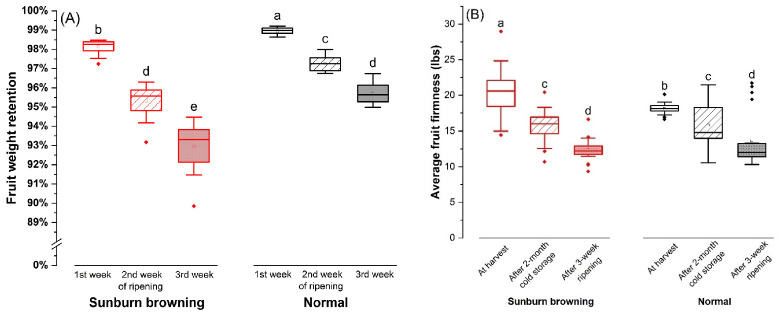
Fruit weight retention and firmness retention of normal and sunburn browning ‘Ambrosia’ apple during storage and ripening: (**A**) weight retention levels during the 1st, 2nd, and 3rd week of ripening at 20 °C; (**B**) loss of fruit firmness after 2-month air storage at 4 °C and subsequent 3-week ripening. Different letters a, b, c, d and e in each subpanel stand for statistical significance at *p* ≤ 0.05 (ANOVA, Tukey pairwise comparison).

**Figure 6 plants-11-01201-f006:**
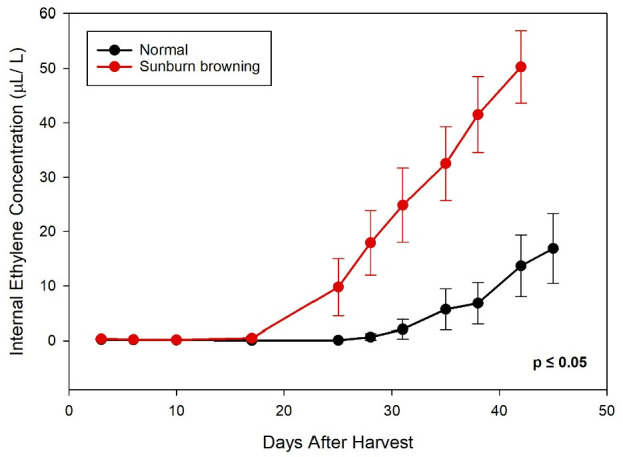
Changes in internal ethylene concentration of normal and sunburn browning ‘Ambrosia’ apples during air storage at 20 °C. Dots and error bars are mean ± standard deviation (*n* = 15).

**Figure 7 plants-11-01201-f007:**
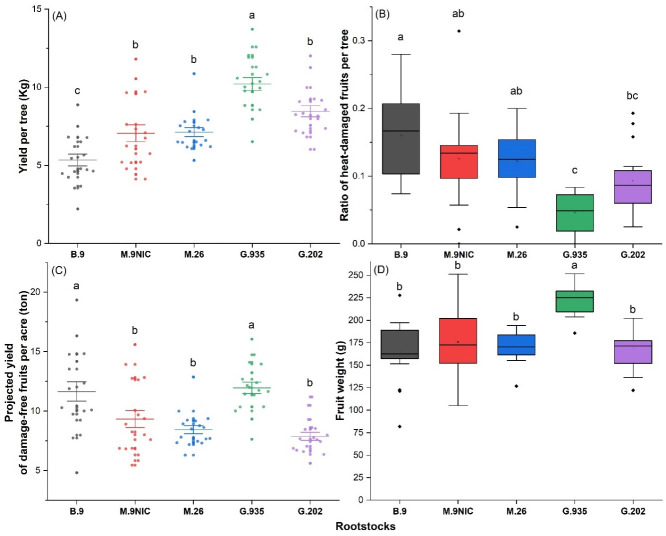
Yield (**A**), ratio of sunburn damaged fruits (**B**), projected damage-free yield per acre (**C**), and average fruit weight (**D**) of ‘Ambrosia’ apple on rootstocks B.9, M.9NIC29, M.26, G.202, and G.935 in 2021. In (**C**), yield was projected based on the estimated optimum tree counts per acre for each rootstock vigor class, i.e., 2180 trees for B.9, 1320 trees for M.9, 1180 trees for M.26 and G.935, and 930 trees for G.202. Different letters in each subpanel stand for statistical significances at *p* ≤ 0.05 (ANOVA, Tukey pairwise comparison).

**Table 1 plants-11-01201-t001:** Fruit weight, soluble solids content (SSC%), malic acid concentration, firmness, dry matter content % (DMC%), and fruit hypanthium water potential (*Ψ*_Fruit_) of normal and sunburn browning ‘Ambrosia’ apple at harvest.

Fruit Condition	Weight (g)	SSC (%)	Malic Acid (mg/100 mL)	Side of Fruit	Firmness (lbs)	DMC (%)	*Ψ*_Fruit_ (MPa)
Normal	188.49 ± 4.14 a	14.23 ± 0.06 b	772.82 ± 17.18 b	Sun-exposed	19.00 ± 0.31 bc	16.99 ± 0.18 b	−1.71 ± 0.07 a
				Shaded	18.68 ± 1.28 c	15.58 ± 0.13 c	−1.68 ± 0.04 a
Sunburn browning	158.09 ± 2.48 b	16.55 ± 0.08 a	1092.89 ± 120.82 a	Sun-exposed	21.22 ± 0.32 a	22.14 ± 0.35 a	−2.47 ± 0.09 b
				Shaded	19.79 ± 1.22 b	15.34 ± 0.20 c	−1.65 ± 0.04 a

Note: Values are mean ± standard errors. Different letters in each column stand for significant differences at *p* ≤ 0.05 (ANOVA, Tukey pairwise comparison).

## Data Availability

Raw data used to generate figures and tables are available on request.

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
