# Peer review of "Characteristics of Sunburn Browning Fruit and Rootstock-Dependent Damage-Free Yield of Ambrosia™ Apple after Sustained Summer Heat Events"

_plants, 2022, doi:10.3390/plants11091201_

Round 1

Reviewer 1 Report

Dear authors,

I want to congratulate you for the work carried out in this project and the very interesting results obtained, which will surely lead to new lines of research in the field of the development of new methodologies to quantify SB in apple trees, as well as the choice of better management, thought rootstock selection in the field to avoid its appearance, as well as the selection of more resistant varieties into breeding programs.

Line 14: Correct punctuation in the enumeration of cell integrity damage

Line 131. Could be desirable describe what kind of stress-related compound was autofluorescent and where was accumulated. Please, describe this figure in more detail in the case of severe damage to tissue

Line 167. Brix (Bx) is expressed in degrees. Soluble solids content (SSC) is expressed in percentage. Instead, 1 ºBx may be equivalent to 1% SSC, but the units are different.

Figures 4 and 5. The font size in X axe is extremely small. please, increase the font size.

Figure 5. The boxplots from the 3rd week of ripening, both normal and sunburn browning, were filled with a highly saturated color which does not allow to discern the average line. please, fill the boxplot with less saturated color.

Figure 7. Why yield per tree and projected yield of damage-free fruits per acre were not represented as a boxplot?

Along with the document, the varieties' names must be between quotation marks.

It should be noted that the data collected only refer to one year and that the experiment should be addressed in future years in order to ensure the robustness of the results obtained. In addition, must be included others parameters as possibly affected by higher temperatures such as the ripening date or the development of red color due to the accumulation of anthocyanins.

In the conclusion, a selection of the best tools for SB measurements and the higher fruit quality trait affected by this damage must be appreciated.

Best regards,

Reviewer

Author Response

Thank you so much for the comments and revision suggestions. Please see our responses in the attachment.

Reviewer 2 Report

Manuscript "Characteristics of sunburn browning fruit and rootstock-dependent damage-free yield of Ambrosia TM apple after sustained summer heat events" presents interesting research results that can be used in agriculture.

Detailed comments:

There are too many keywords - please choose the most important ones.

The results are correctly described, please review the units and adjust to the journal requirements (eg line 209).

The authors presented a small discussion of the results with other works, this could be supplemented.

There is no unequivocal summary of the research in the manuscript.

Only about 32% of the cited literature is from the last 5 years - authors should supplement the mansukrypt with the latest research.

Author Response

Thank you so much for the comments and suggestions. Please kindly review our responses in the attachment. 

Reviewer 3 Report

Summary:  

The authors provide a timely study of the effect of sunburn on apples growing during a heatwave in Canada.  The methods are appropriate and they report an interesting mitigative effect of different root stocks that is worthy of further investigation as an effort to mitigate future stress due to climate changes.  Some attention to details needs to be made as outline din comments and also Fig 4A was omitted from the uploaded manuscript although I am trusting that they did do the experiment.

Major and Minor Comments

Introduction:  OK

Results:

Fig 1 OK but it would be helpful to indicate what were the harvest days of the fruits shown in B ad C on the temperature chart.  Also in Fig 1C is the fruit on the right meant to be mild?  or should it be labelled as in Fig 1B.

Fig 2. In Figure 2 is it accurate to say that the cells in severely damaged tissue are no longer intact and that chloroplasts are lysed? - this distributing chlorophyll throughput the cytoplasm? - explaining what the highly fluorescent compounds may be?  or any better attempt to identify what the autofluorescent material is?

Fig 3. In the images shown the cuticle is hard to see in the moderately damaged tissue and in fact the cuticle seems thickest in the non-damaged tissue.  Maybe they can provide images that are clearer and more convincing of the thickness reported in Fig 3B.

Fig 4: Fig4A is missing!   Fig4B is OK.

Table 1:  OK

Fig 5: Clearly shows that SB fruit are losing water and therefore the weight is dropping faster than normal fruit - also firmness is reduced in SB fruit at harvest time.

Fig 6:  OK.

Fig 7: This is probably the most useful output of the experiments as it shows that different root stocks resulted in different yields and less damage - when projected onto yield per acre this can have significant savings. However I am puzzled as to how stock B.9 which has the lowest yield per acre (Fig7A) and the highest ration of damaged fruits per tree  (Fig 7B) has the highest projected yield of damage free fruits per acre? is that because more trees are planted per acre?

Discussion:

In the first paragraph I am not very convinced about the effect of cuticle thickness as the images underlying Fig 3B are not very convincing - especially the moderately affected fruits as mentioned above - maybe they can provide more convincing images.  I am not convinced that the cuticle serves to reduce surface temperature - more likely it has to do with reduced water loss (or increased water loss when it is damaged).

The use of different stains could have been used to try and  identify or at least narrow down the autofluorescent material accumulating in the damaged cells

Peter Kitin, Satoshi Nakaba, Christopher G Hunt, Sierin Lim, Ryo Funada, Direct fluorescence imaging of lignocellulosic and suberized cell walls in roots and stems, AoB PLANTS, Volume 12, Issue 4, August 2020, plaa032, https://doi.org/10.1093/aobpla/plaa032

Phenolic accumulation is also known to accumulate or be enhance under high light conditions which likely include enhanced exposure to UV light (but not measured in this study).

Surjadinata, Bernadeth B et al. “UVA, UVB and UVC Light Enhances the Biosynthesis of Phenolic Antioxidants in Fresh-Cut Carrot through a Synergistic Effect with Wounding.” Molecules (Basel, Switzerland) vol. 22,4 668. 24 Apr. 2017, doi:10.3390/molecules22040668

I found the study on different root stocks to be the most valuable and applied aspect of the manuscript and should prove useful as climate conditions force horticulturalists to mitigate the increased damage without access to more water.

Materials and Methods:

I found this section to be quite detailed.

References:  OK

Author Response

(The authors gave the same response as above.)
